# Microclimatic Tipping Points at the Beech–Oak Ecotone in the Western Romanian Carpathians

**Stefan Hohnwald [1,2], Adrian Indreica [2], Helge Walentowski [1,\*]  and Christoph Leuschner [3]**

[1]  Faculty of Resource Management, HAWK University of Applied Sciences and Arts, Büsgenweg 1a, 37077 Göttingen, Germany; stefan.hohnwald@hawk.de

[2]  Department of Silviculture, Transilvania University of Braşov, Șirul Beethoven no. 1, 500123 Brasov, Romania; adrianindreica@unitbv.ro

[3]  Plant Ecology, Albrecht von Haller Institute for Plant Sciences, University of Goettingen, Untere Karspüle 2, 37073 Göttingen, Germany; cleusch@gwdg.de

\*  Correspondence: helge.walentowski@hawk.de; Tel.: +49-551-5032-177

**Abstract:** European beech (*Fagus sylvatica*) is a drought-sensitive species that likely will retreat from its xeric distribution edges in the course of climate warming. Physiological measurements indicate that the species may not only be sensitive to soil water deficits, but also to high temperatures and elevated atmospheric vapor pressure deficits (vpd). Through microclimatological measurements in the stand interior across near-natural beech forest–oak forest ecotones, we searched for microclimatic tipping points in the contact zone with the aim to define the thermic and hydrometeorological limits of beech more precisely. In three transects in the foothills of the Romanian western Carpathians, we measured in mid-summer 2019 air temperature, relative air humidity, and vpd at 2 m height in the stand interior across the ecotone from pure oak to pure beech forests, and compared the readings to the microclimate in forest gaps. Mean daytime temperature (T) and vpd were by 2 K and 2 hPa, respectively, higher in the oak forests than the beech forests; the extremes differed even more. Especially in the second half of the day, the oak forests heated up and were more xeric than the beech forests. Part of the differences is explained by the elevation difference between oak and beech forests (200–300 m), but species differences in canopy structure, leaf area, and canopy transmissivity enhance the microclimatic contrast. Our T and vpd data point to thresholds at about 30 °C and 25 hPa as maxima tolerated by beech in the lowermost shade canopy for extended periods. In conclusion, the rather sharp stand microclimatic gradient demonstrated here for the xeric distribution limit of beech may well be the decisive factor that hinders the spread of beech into the warmer oak forests.

**Keywords:**  air temperature; *Fagus sylvatica*; *Quercus frainetto*; *Quercus petraea*; rear edge; stand microclimate; thermal limit; vapor pressure deficit

## 1. Introduction

The extraordinary drought and heat in the summers 2018 and 2019 has demonstrated the climatic vulnerability of European beech (*Fagus sylvatica* L.) in many parts of its Central European distribution range [1,2]. This iconic tree species dominates large parts of Central Europe's natural forest vegetation due to its high competitiveness in moist and cool temperate climates, and it is an important timber tree in many countries [3,4]. At its southern and south-eastern range edges, beech is limited most likely by summer drought and probably also heat. In southern and south-eastern Europe, beech occurs only at higher elevations in the mountains, thus avoiding the more summer dry and hot lowland regions, where the beech forests are replaced by oak-rich sub-Mediterranean forest communities of the Quercetalia pubescenti-petraeae (thermophilic mixed oak forests) and Carpinetalia betuli (oak-hornbeam forests)

orders [5–7]. In Central Europe, beech forests are replaced by oak-rich forests in certain lowland regions with sub-continental or sub-Mediterranean climates and annual precipitation < c. 600 mm [4]. Various indices have been proposed to characterize the transition between beech-dominated and oak-dominated forests at the xeric distribution limit of beech, among them the Ellenberg quotient (EQ, average July air temperature × 1000 divided by annual precipitation) [8–11]. There is some evidence that beech is unable to maintain its large and shading leaf area beyond a climatic threshold, which is characterized by an EQ of ≈30, and thus to successfully suppress its competitors [1,12]. This limit has been termed the climatic turning point for beech and oak in Central Europe [13]. Most likely this is not a sharp line, but a continuous ecotone between beech-dominated and oak-dominated forests.

Climate change scenarios predict for western Central Europe an increase in annual mean temperature by 1.6–3.8 °C in the next 60 years [14], which should shift the border line separating natural beech and thermophilous oak forests toward higher elevations and to regions with higher precipitation, as elevated temperatures increase the evaporative demand of the atmosphere, thereby deteriorating the climatic water balance. Biologically even more influential than rising mean temperatures is the expected increase in frequency and severity of extreme heat and drought events [15–19], which may lower the vitality and rise the mortality of beech in many regions. Foresters planning the species composition of production forests for the 21st century thus are facing difficult decisions, as the exact thermal and hydrometeorological limits of beech are not well known. Physiological investigations indicate that beech may be limited in a warmer and drier climate by several factors, notably water deficits, but also heat and elevated vapor pressure deficits (vpd), or a combination of these factors [20–24]. Foresters and tree ecophysiologists would greatly profit from a more precise knowledge of maximum temperatures and vapor pressure deficits tolerated by beech at its distribution limits, as it could help defining habitats suitable for beech planting more precisely [25,26].

As most Central European forests are influenced by forest management [27,28], the natural ecotone between beech and oak forests and the associated microclimatic conditions can hardly be studied in the core of the beech distribution range. This is possible in the Balkans, where forest use in many regions is less intense and near-natural beech and oak forests are still present in places. Here, we present the results of a microclimatic gradient study in three near-natural oak–beech forest ecotones in the western Romanian Carpathians, which aimed at identifying maximum air temperatures and vpd levels that are tolerated by beech at its xeric distribution limit. In this region, beech forests with high dominance of *F. sylvatica* at higher elevation border to xerothermic mixed oak forests at lower elevation, the latter consisting of *Quercus cerris* L., *Quercus petraea* var. *dalechampii* (Ten.) Cristur., and *Quercus frainetto* Ten., among other broadleaf species [29]. The climate of this region in the west of South-east Europe at the trailing edge of beech distribution is currently 2–3 K warmer than at submontane elevation in the core of Central Europe's beech forest region in Germany, while being equivalent to the climatic conditions predicted for Central Europe for the mid and late 21st century according to regional climate models [29–31]. This offers the opportunity to study expected future growing conditions of beech in Central Europe by means of a space-for-time substitution approach and to localize current thermal and hydrometeorological limits of the species.

In the three transects across the oak–beech forest ecotone, temperature, and air humidity sensors with logging function were placed in mid-summer 2019 in the stand interior and the data compared to sensor readings taken in the open outside the forest. As oak forests occur on the foothills of the Romanian western Carpathians at colline elevation (<300/400 m a.s.l.) and beech forests higher upslope at submontane to montane elevation (>500/600 m a.s.l.) [6,7,29], the transects span an elevation distance of 200–300 m, equivalent to a temperature decrease of 1.1–1.8 K. This has to be taken into account in the data analysis. To characterize the stand microclimate in the transition zone between oak and beech forest, two independent measuring points were selected in the oak/beech ecotone of each transect at an elevation distance of about 100 m to each other. An additional measuring point was installed in each transect in a north-facing beech forest at about mid-slope elevation. These isolated beech stands exist

as outposts at lower elevation in a mostly oak-dominated woodland and may give valuable additional information about the microclimatic requirements of beech.

The climatic conditions inside a forest are the result of the regional climate (which is influenced by elevation, slope inclination, and aspect) and the modifying effect of the canopy as an expression of the species influence. Measurements in the forest interior at 2 m height were assumed to characterize the growing conditions of tree regeneration and the trees' shade foliage, while the microclimatic conditions of the sun foliage should be more similar to the readings of the station in the open.

We asked the following four questions: (i) How much warmer is the oak forest interior on average and how much drier the air as compared to the beech forest interior in the contact zone of both communities? (ii) How much do temperature and vpd maxima (and relative air humidity minima) differ between the interior of beech and oak forests? (iii) Are the microclimatic differences driven by high insolation during cloudless summer days or do they exist independent of weather conditions? (iv) Can the effect of stand structure (beech vs. oak forest) on the microclimate be separated from the influence of local climatic differences along the slope?

## 2. Materials and Methods

### 2.1. Study Sites and Microclimate Measurements

Stand microclimate measurements were conducted in summer 2019 (15 June to 27 July) in three transects spanning from pure oak forest through an oak/beech ecotone to pure beech forest at colline to lower montane elevation in the south-western Carpathians in western Romania. The transects were selected on slopes with south-western exposition (except for the additional beech stands on northern slopes: Be(N)). Every transect consisted of five measuring points, pure oak forest (Oa; *Quercus petraea*, *Q. cerris*, *Q. frainetto*), two measuring points in the oak/beech forest ecotone (lower and upper site; Oa/Be(lo) and Oa/Be(up)), and pure beech forest (Be), all at south-western exposition, and an additional beech forest on northern exposition (Be(N)) for comparison. A sixth measuring point was established outside the forest in a large gap (Ga) at about mid-slope position. The transects Milova (46.1° N/21.8° E) and Maciova (45.5° N/22.2° E) are located north-east and south-east of Timisoara, the transect Eşelniţa (44.7° N/22.3° E) west of Orşova close to river Danube (Figure 1). As the transects were designed to serve as replicates on the landscape level, they were selected for sufficient comparability in terms of stand structure and tree species composition, exposition, soil types, and overall climatic conditions. All stands were mature forests of about 25–35 m height with closed canopy and had been managed only at low intensity. In the oak and beech forests, oak species and beech, respectively, each contributed with at least 85% to the stems, while remaining stems belonged to accompanying species such as *Carpinus betulus* L., *Acer campestre* L. and *Tilia tomentosa* Moench. In the ecotone, the oak species (*Q. petraea*, *Q. cerris*, *Q. frainetto*) and beech each contributed with about 30% to the basal area, while the remainder belonged mostly to *Carpinus* and the *Tilia* species.

The climate of the study region is cool-temperate sub-continental with warm summers and relatively cold winters (Table 1). The lapse rate of annual precipitation was calculated with +45 mm/100 m, the temperature lapse rate with −0.5 K/100 m [32]. All forests stock on acidic bedrock, which at many places is covered by a loess layer of up to 50 cm depth. Soil types are (eutric) Cambisols and Luvisols.

Air temperature, relative air humidity (RH), and atmospheric saturation deficit (vpd) were recorded at 15-min intervals with ibutton DS1923 Hygrochron sensors (Measurement Systems (MSL) Ltd., Berkshire, UK) that stored the data with integrated loggers. The sensor accuracy is 0.5 °C for temperature with a resolution of 0.0625 K, and a 0.5% accuracy for RH [33]. The sensors were fixed at 2 m height inside the forests to the bark of a trunk and were shielded against direct radiation with white plastic cups of 0.4 L volume placed upside down over the sensor to allow for sufficient air turbulence. In the ecotones, sensors were placed in forest patches with equal abundance of oak and beech trees. All measuring points were located at least 100 m distant to the next forest edge. The sensors in the

gaps were fixed on poles at 2 m height above ground and were shielded against radiation as described above. The gaps had a diameter of at least 20 m (Table 2).

**Table 1.** Climate data for the transects in Maciova, Milova, and Eşelniţa according to climate-data.org [34] using the nearest available localities.

| Transect | Unit | Milova | Maciova | Eşelniţa |
|---|---|---|---|---|
| Nearest localities | | Lipova (126 m) | Mâtnicu Mare (181 m) | Bârza (98 m) |
| Mean annual temperature | °C | 10.8 | 10.7 | 11.1 |
| Minimum temperature of coldest month | °C | −4.5 | −4.0 | −3.3 |
| Mean temperature of warmest quarter | °C | 20.3 | 20.0 | 20.7 |
| Mean annual precipitation | mm | 604 | 639 | 621 |
| Precipitation of warmest quarter | mm | 201 | 218 | 202 |

**Table 2.** Location of the five measuring points in the three transects and in the forest gaps with forest type, acronym, elevation, length of measuring periods (full day: 0:00–23:45), and the Ellenberg quotient (EQ). EQ was calculated based on lapse rates of −0.5 K/100 m and +45 mm/100 m, according to [32], applied to the nearest available weather stations included in climate-data.org ([34]; Table 2).

| Measuring Points | Acronym | Eleva-tion a. s. l. (m) | Measuring Period 2019 | Number of Full Days | EQ |
|---|---|---|---|---|---|
| **Transect 1—Milova** | | | | | |
| Oak forest | Oa | 298 | 19.6.–23.7. | 35 | 29.0 |
| Oak/beech ecotone (lower) | Oa/Be(lo) | 407 | 20.6.–24.7. | 35 | 26.7 |
| Oak/beech ecotone (upper) | Oa/Be(up) | 487 | 20.6.–24.7. | 35 | 24.9 |
| Beech forest | Be | 504 | 20.6.–24.7. | 35 | 24.6 |
| Beech forest N-slope | Be(N) | 412 | 20.6.–24.7. | 35 | |
| Gap | Ga | 234 | 20.6.–23.7. | 34 | |
| **Transect 2—Maciova** | | | | | |
| Oak forest | Oa | 334 | 18.6.–22.7. | 35 | 28.3 |
| Oak/beech ecotone (lower) | Oa/Be(lo) | 450 | 17.6.–22.7. | 36 | 25.6 |
| Oak/beech ecotone (upper) | Oa/Be(up) | 594 | 18.6.–22.7. | 35 | 22.7 |
| Beech forest | Be | 593 | 18.6.–22.7. | 35 | 22.7 |
| Beech forest N-slope | Be(N) | 359 | 21.6.–22.7. | 32 | |
| Gap | Ga | 370 | 18.6.–22.7. | 35 | |
| **Transect 3—Eşelniţa** | | | | | |
| Oak forest | Oa | 563 | 16.6.–27.7. | 42 | 23.1 |
| Oak/beech ecotone (lower) | Oa/Be(lo) | 623 | 15.6.–26.7. | 42 | 22.0 |
| Oak/beech ecotone (upper) | Oa/Be(up) | 657 | 16.6.–26.7. | 41 | 21.4 |
| Beech forest | Be | 877 | 16.6.–26.7. | 41 | 18.1 |
| Beech forest N-slope | Be(N) | 601 | 15.6.–26.7. | 42 | |
| Gap | Ga | 580 | 15.6.–26.7. | 42 | |

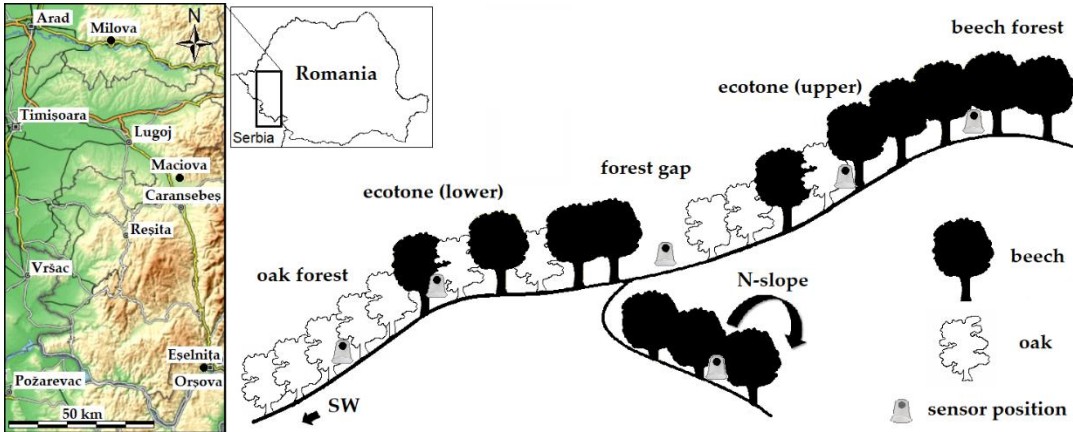

**Figure 1.** Study region in the western Romanian Carpathians (source OpenTopoMap, licence CC-BY-SA, slightly altered), and scheme of the transects from oak to beech forest on a south-west exposed slope. The location of microclimatic measuring points (sensor position) is indicated. Additional measuring points were established in a north-facing beech forest at mid-slope elevation and in a forest gap.

## 2.2. Data Analysis

For the calculation of vpd, we used the Teten formula which gives very similar results as more sophisticated formulas and is recommended for its convenience [35,36]. Accordingly, vpd was calculated from measured air temperature using the expression

$$e_s\,(T) = 611 \times 10^{\,(T/(T\,+\,237.3)\,\times\,17.27)} \tag{1}$$

to obtain saturating vapor pressure ($e_s$) at temperature T, which was multiplied with RH (in %) to give actual vapor pressure ($e_a$); vpd is the difference between $e_s$ and $e_a$. From the measured 15-min values, daily means, and daily maxima (T, vpd) and minima (RH) were calculated, either for the whole measuring period or only for cloudless days, and for the whole day (00:00–23:45 h) or only the daytime period, covering roughly the period between dawn and dusk (09:00–21:00 h). As cloudless days, we counted all days without rain and less than 50% cloud cover. These days comprise the 20 hottest days of the measuring period. To exclude effects of varying day length on the calculation of means, we fixed the daytime period arbitrarily to 9 a.m. to 9 p.m.

The data were stored in a Microsoft Excel database, and first daily means with standard errors were calculated for every measuring point, and subsequently means by pooling the data of the corresponding measuring points of the three transects. Overall means were then calculated by averaging over all days of the measuring period, over all cloudless days, or by averaging over all daily extreme values (maxima or minima). In addition, the vpd values recorded between 9 a.m. and 9 p.m. were summed up over 19 cloudless days at all measuring stations to obtain a 'cumulative saturation deficit' for each station (2793 individual measurements, expressed in MPa) as a measure of the local long-term evaporative demand.

To rank the five forest types and the gap measuring points with respect to their temperature and vpd maxima and RH minima, we selected 19 different microclimatic variables (different means and maxima/minima) and calculated a simple score by assigning for every variable a rank from 1 to 6 to each forest type. The individual rank scores of a forest type or the gap were then averaged over the 19 variables to give a mean score, with the lowest mean score characterizing the hottest and driest and the highest score the coolest and moistest measuring point.

Analysis of variance (ANOVA) was applied to test for significant differences between means of the five measuring points, using the *aov()* procedure of the software platform R [37]. If at least one population mean was different from others with respect to the main effects and interaction of the underlying sample, comparisons between all single means of the five measuring points were conducted

with subsequent multiple post-hoc testing, using Tukey's HSD test, implemented in the *aov()* procedure of R. Linear models were calculated to examine the influence of the categorial variables measuring point (five forest types and gap), transect (three transects), and the measuring point × transect interaction on the means of temperature, RH, and vpd, and their extremes, that were also considered in the ANOVAs. The residuals were inspected for normal distribution and equality of variances. A significance level of $p < 0.05$ was used throughout the study. To illustrate the distribution of raw data, some exemplary boxplots of temperature, air humidity, and vpd data are provided in the attachment (Appendix A Figures A1–A3).

## 3. Results

The diurnal course of air temperature on three cloudless days in July in the five forest types and the gap of the Milova transect demonstrates a more rapid heating up of the oak stand interior in the morning hours, a by 3 K warmer stand around noon, and a longer heat storage in the afternoon, as compared to the beech stand (Figure 2). During night, in contrast, the beech stand was by 2 K warmer than the oak stand. The two measuring points in the oak/beech ecotone held intermediate positions with respect to the diurnal temperature fluctuations. Thermal differences between stand types were less pronounced during cloudy and rainy days.

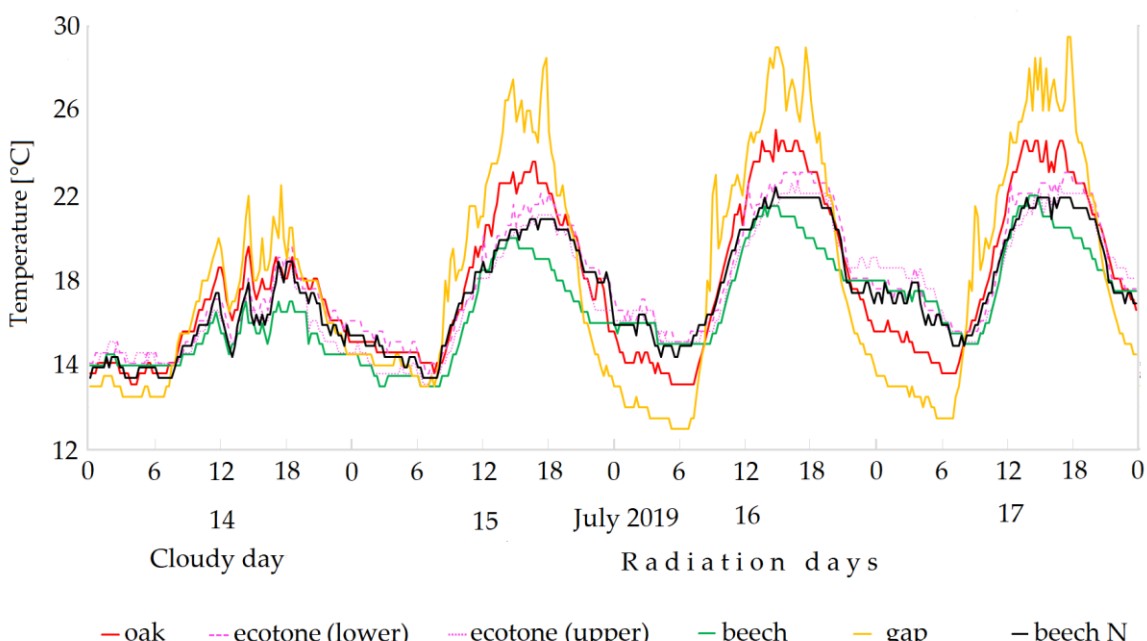

**Figure 2.** Diurnal course of air temperature at the six measuring points of the Milova transect on a cloudy and three cloudless days (radiation days) in July 2019.

Figure 3 shows typical examples of diurnal courses of air temperature, air humidity, and vpd for the gap station in the Maciova transect. The five days include the highest vpd value (41.6 hPa on July 3) recorded in whole measurement period, when air temperature reached almost 40 °C and RH dropped to below 20%.

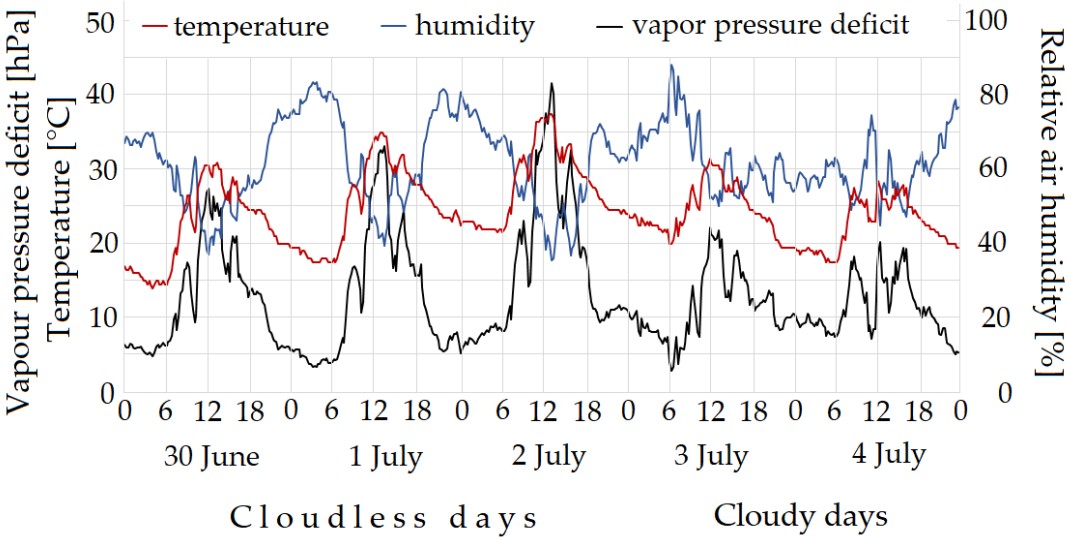

**Figure 3.** Diurnal courses of air temperature, relative air humidity, and vapor pressure deficits (vpd) on three cloudless days and two cloudy days in the gap of the Maciova transect.

Along the three transects on slopes with south-western exposition, air temperature in the stand interior decreased significantly from the oak forests through the oak/beech ecotone toward the beech forests by 1.2 K, when averaged over the whole measuring period (35–42 days), and by 2.0 K when only the daytime period was considered (Table 3). The gradient in the three transects was similar for cloudless days (1.9 K), but steeper when the daily maxima (2.9 K) and the absolute maxima recorded in the whole period (3.3 K) were considered. In the beech forests on the northern slope, which were located at elevations similar to the Oa or Oa/Be(lo) sites, temperature averages were only slightly lower than in the south-west exposed oak forests (difference 0.2 K for the whole period, and 0.6 K for the daytime period); the difference in absolute maxima was greater (1.6 K). All forest sites were on average cooler than the measuring points in the open (Ga); the difference was particularly great on cloudless days (by 3.7–5.6 K lower daily maxima). The absolute temperature maxima in the gaps were up to 6.5 K higher than in the forest interior and reached 37.4 °C as the temperature extreme on July 2 in the Maciova transect (Table 3).

**Table 3.** Mean air temperatures (in °C), mean daytime temperatures, mean daily maxima, and mean absolute maxima (with SE in brackets) at the five measuring points in the forest and in the gap, averaged over the three transects. The daytime period refers to 9 a.m.–9 p.m. Significantly different means are marked with different small letters. The first four columns refer to all days of the measuring period, the last two columns only to days with less than 50% cloudiness and no rain (cloudless). N gives the number of individual 15-min values used for averaging.

| N | Mean of Whole Period 10,672 | Daytime Mean 6168 | Mean Daily Maximum 112 | Absolute Maximum 3 | Daytime Mean (Cloudless) 2939 | Mean Daily Maximum (Cloudless) 60 |
|---|---|---|---|---|---|---|
| Oa | 20.7 (0.04) [a] | 22.7 (0.04) [a] | 26.0 (0.27) [a] | 31.8 (0.17) [a] | 23.9 (0.06) [a] | 27.0 (0.34) [a] |
| Oa/Be(lo) | 20.6 (0.03) [a] | 22.0 (0.04) [b] | 24.6 (0.26) [b,e] | 30.3 (0.69) [a,b] | 23.3 (0.06) [b] | 25.6 (0.32) [a,b,c] |
| Oa/Be(up) | 20.4 (0.03) [b] | 21.7 (0.04) [c] | 24.1 (0.26) [b] | 29.4 (0.35) [b] | 22.8 (0.06) [c] | 25.1 (0.34) [b,c] |
| Be | 19.5 (0.03) [c] | 20.7 (0.04) [d] | 23.1 (0.26) [c] | 28.5 (0.06) [b] | 22.0 (0.06) [d] | 24.1 (0.32) [c] |
| Be(N) | 20.5 (0.03) [b] | 22.1 (0.04) [b] | 24.8 (0.27) [e] | 30.2 (0.46) [a,b] | 23.3 (0.05) [b] | 25.8 (0.36) [a,b] |
| Ga | 21.2 (0.05) [d] | 23.8 (0.05) [e] | 29.1 (0.33) [d] | 35.0 (1.56) [a,b] | 25.7 (0.07) [e] | 30.7 (0.39) [d] |

Relative air humidity showed the lowest mean value in the upper oak/beech ecotone (Oa/Be(up)) with 68.9%, and not in the oak forests (72.7%). The oak mean was even slightly higher than the

mean of the beech forests (71.6%; Table 4). This sequence of sites changed when daytime means were considered, which were highest in the beech forests. In correspondence, daily mean and absolute RH minima reached their lowest values in the oak forests. A similar gradient from the beech to the oak forests was observed on cloudless days, but the values were on average by about three percentage points lower than when all days were considered. Absolute minima were six percentage points lower in the oak than the beech forests. The beech forests on northern slopes reached lower absolute RH minima than the beech forests higher up on south-western slopes, but their long-term means were more similar. In the gaps, RH reached the lowest minima and lowest daytime means of all measuring points, while the RH mean of the whole period was very similar to the means measured inside the forests (Table 4).

**Table 4.** Mean relative air humidity (RH, in %), mean daytime RH, mean daily RH minimum, and mean absolute minimum (with SE in brackets) at the five measuring points in the forest and in the gap, averaged over the three transects. The daytime period refers to 9 a.m.–9 p.m. Significantly different means are marked with different small letters. The first four columns refer to all days of the measuring period, the last two columns only to days with less than 50% cloudiness and no rain (cloudless). N gives the number of individual 15-min values used for averaging.

| N | Mean of Whole Period 10,672 | Daytime Mean 6168 | Mean Daily Minimum 112 | Absolute Minimum 3 | Daytime Mean (Cloudless) 2939 | Mean daily Minimum (Cloudless) 60 |
|---|---|---|---|---|---|---|
| Oa | 72.7 (0.14) [a] | 66.8 (0.18) [a] | 52.7 (1.04) [a] | 34.8 (1.85) [a,b,c] | 64.2 (0.26) [a] | 49.6 (0.34) [a] |
| Oa/Be(lo) | 70.7 (0.14) [b] | 67.4 (0.17) [b] | 55.3 (1.00) [b] | 38.4 (1.10) [a,c] | 65.1 (0.24) [b] | 52.3 (0.32) [a,b] |
| Oa/Be(up) | 68.9 (0.13) [c] | 66.5 (0.17) [a] | 54.6 (1.02) [a,b] | 37.2 (0.75) [a,b] | 65.7 (0.25) [b,e] | 53.5 (0.34) [b] |
| Be | 71.6 (0.13) [d] | 69.5 (0.15) [c] | 58.0 (0.99) [b] | 40.9 (0.35) [c] | 67.0 (0.22) [c] | 54.8 (0.32) [b] |
| Be(N) | 71.5 (0.13) [d] | 67.8 (0.16) [e] | 56.2 (1.01) [b] | 38.0 (1.50) [a,b,c] | 65.8 (0.23) [e] | 53.1 (0.36) [b] |
| Ga | 70.8 (0.16) [b] | 63.2 (0.18) [d] | 46.1 (0.88) [c] | 32.0 (0.98) [b] | 58.8 (0.25) [d] | 42.9 (0.39) [c] |

Contrary to expectation, mean vpd was highest in the oak/beech ecotone and not in the oak forests with 11.2 and 8.0 hPa at the Oa/Be(lo) and Oa/Br(up) measuring points, while vpd averaged at 7.4 hPa in the oak forests (Table 5). Similarly, mean daily maxima were higher at the Oa/Be(lo) measuring point than in Oa (16.6 vs. 14.7 hPa). In contrast, absolute vpd maxima were highest in the oak forests with 27.3 hPa. The single most extreme vpd value recorded in the beech forests was 21.0 hPa (on 21 July, at 18:40), while in the oak forests 29.3 hPa was reached as the extreme. The highest vpd of the study was recorded in the gap of Maciova on 2 July (Figure 3). The 15-min vpd values summed up over the whole daytime periods of 19 cloudless days showed that the cumulative saturation deficit was 20% greater in the oak forests, and even 34% greater in the ecotone (Oa/Be(lo)), than in the beech forests (Table 5, last column). The north-exposed beech forests at mid elevation had a higher cumulative vpd than the beech forests on the SW-exposed slope higher up, but a lower one than the ecotone at similar elevation.

The ranking of the five forest types and the gap with respect to microclimatic extremes revealed the hottest and driest conditions for the oak forests, and only a small difference to the lower oak/beech ecotone, while the upper ecotone had a markedly higher score (Table 6). Accordingly, the coolest and moistest forest type were the beech forests on the south-western slope, while the beech forests on the northern slope (at lower elevation) had a score only slightly higher than the Oa/Be(up) measuring point. Not surprisingly, the gap had a by far lower score than all five measuring points in the forest.

**Table 5.** Mean atmospheric saturation deficit (vpd, in hPa), mean daytime vpd, mean daily vpd maximum, mean absolute maximum vpd, and vpd sums (with SE in brackets) at the five measuring points in the forest and in the gap, averaged over the three transects. The daytime period refers to 9 a.m.–9 p.m. Significantly different means are marked with different small letters. The first four columns refer to all days of the measuring period, the last two columns only to days with less than 50% cloudiness and no rain (cloudless). N gives the number of individual 15-min values used for averaging. The vpd sums given in the last column were calculated by adding the 15-min values of 19 cloudless days (unit: MPa).

| N | Mean of Whole Period 10,672 | Daytime Mean 5398 | Mean Daily Maximum 111 | Absolute Maximum 3 | Daytime Mean (Cloudless) 2939 | Mean Daily Maximum (Cloudless) 59 | vpd Sums of Daytime Period (Cloudless) |
|---|---|---|---|---|---|---|---|
| Oa | 7.4 (0.05) [a] | 9.9 (0.07) [a] | 14.7 (0,48) [a] | 27.3 (1.33) [a] | 11.3 (0.10) [a] | 17.6 (0.57) [a] | 103.5 (0.14) [a] |
| Oa/Be(lo) | 11.2 (0.06) [b] | 12.6 (0.08) [b] | 16.6 (0.50) [b] | 23.7 (1.85) [a,c] | 13.5 (0.11) [b] | 17.7 (0.61) [a] | 125.6 (0.89) [a,b] |
| Oa/Be(up) | 8.0 (0.04) [c] | 9.2 (0.06) [c] | 13.5 (0.41) [a,e] | 22.2 (0.23) [a,c] | 10.4 (0.08) [c] | 15.0 (0.51) [b] | 95.9 (0.65) [a,b] |
| Be | 6.8 (0.04) [d] | 7.8 (0.05) [d] | 11.8 (0.36) [c] | 19.9 (0.69) [b,c,d] | 9.1 (0.07) [d] | 13.2 (0.44) [c] | 83.3 (0.14) [b] |
| Be(N) | 7.3 (0.04) [a] | 9.0 (0.06) [e] | 13.5 (0.37) [e] | 21.1 (0.98) [b,c,d] | 9.8 (0.07) [f] | 14.6 (0.43) [b] | 90.0 (0.13) [a,b] |
| Ga | 8.6 (0.06) [e] | 12.5 (0.09) [b] | 22.1 (0.63) [d] | 35.1 (4.50) [ad] | 14.5 (0.13) [e] | 25.0 (0.77) [d] | 134.0 (0.45) [a] |

**Table 6.** Ranking of the five forest types and the gap according to microclimatic means and extremes recorded. In total, 19 different variables related to air temperature means and maxima, RH means and minima, and vpd means and maxima were expressed for the six measuring points with scores from 1 to 6 (1 = hottest and driest, 6 = coolest and moistest) and a mean score calculated by averaging over the 19 scores.

| Variable | Oa | Oa/Be(lo) | Oa/Be(up) | Be | Be(N) | Ga |
|---|---|---|---|---|---|---|
| T-mean of whole period | 2 | 3 | 5 | 6 | 4 | 1 |
| T-daytime mean | 2 | 4 | 5 | 6 | 3 | 1 |
| T-daily maximum | 2 | 4 | 5 | 6 | 3 | 1 |
| T-absolute maximum | 2 | 3 | 5 | 6 | 4 | 1 |
| T-daytime mean (cloudless) | 2 | 3 | 5 | 6 | 4 | 1 |
| T-daily maximum (cloudless) | 2 | 4 | 5 | 6 | 3 | 1 |
| RH-mean of whole period | 6 | 2 | 1 | 5 | 4 | 3 |
| RH-daytime mean | 3 | 4 | 2 | 6 | 5 | 1 |
| RH-daily minimum | 2 | 4 | 3 | 6 | 5 | 1 |
| RH-absolute minimum | 2 | 5 | 3 | 6 | 4 | 1 |
| RH-daytime mean (cloudless) | 2 | 3 | 4 | 6 | 5 | 1 |
| RH-daily minimum (cloudless) | 2 | 3 | 5 | 6 | 4 | 1 |
| vpd-mean of whole period | 4 | 1 | 3 | 6 | 5 | 2 |
| vpd-daytime mean | 3 | 1 | 4 | 6 | 5 | 2 |
| vpd-daily maximum | 3 | 2 | 4 | 6 | 5 | 1 |
| vpd-absolute minimum | 2 | 3 | 4 | 6 | 5 | 1 |
| vpd-daytime mean (cloudless) | 3 | 2 | 4 | 6 | 5 | 1 |
| vpd-daily maximum (cloudless) | 3 | 2 | 4 | 6 | 5 | 1 |
| vpd-sum of cloudless days | 3 | 2 | 4 | 6 | 5 | 1 |
| Mean score | 2.6 | 2.8 | 3.9 | 5.9 | 4.4 | 1.2 |

The thermoisopleth diagrams of 13 days for the oak and beech forest in Eşelniţa show that the warming of the stand interior is beginning not before 10 a.m. and reaches its peak regularly at 4 p.m. (Figure 4). The higher warmth during afternoon hours is visible not only on cloudless hot days but also on overcast cooler days. Beech and oak forests differ even more with respect to vpd, which can be high not only around 4 p.m. but also during certain nights during dry periods (Figure 5). The differential graphs in Figure 6 show that the oak forest is warmer by 2 K than the beech forest most of the day, and during afternoons and in some nights even by 4–6 K. The oak forest has a by 4–6 hPa higher vpd

during most of the afternoon periods, while during late night and the first half of the day, vpd can be higher or lower than in the beech forest.

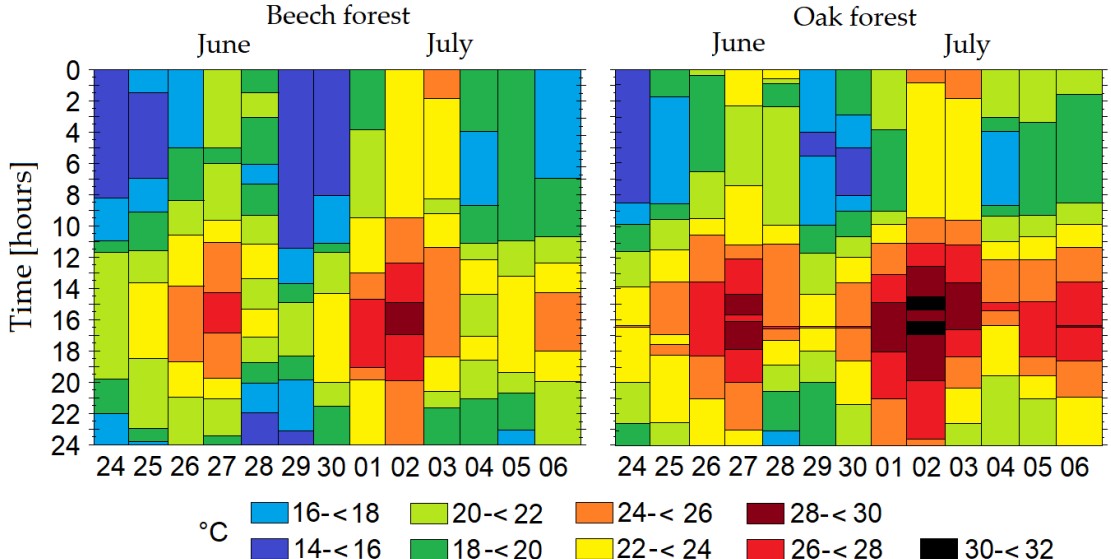

**Figure 4.** Diurnal variation in air temperature on 13 days in summer 2019 (24 June—6 July) in the interior of the beech forest (**left**) and oak forest (**right**) of the Eşelniţa transect.

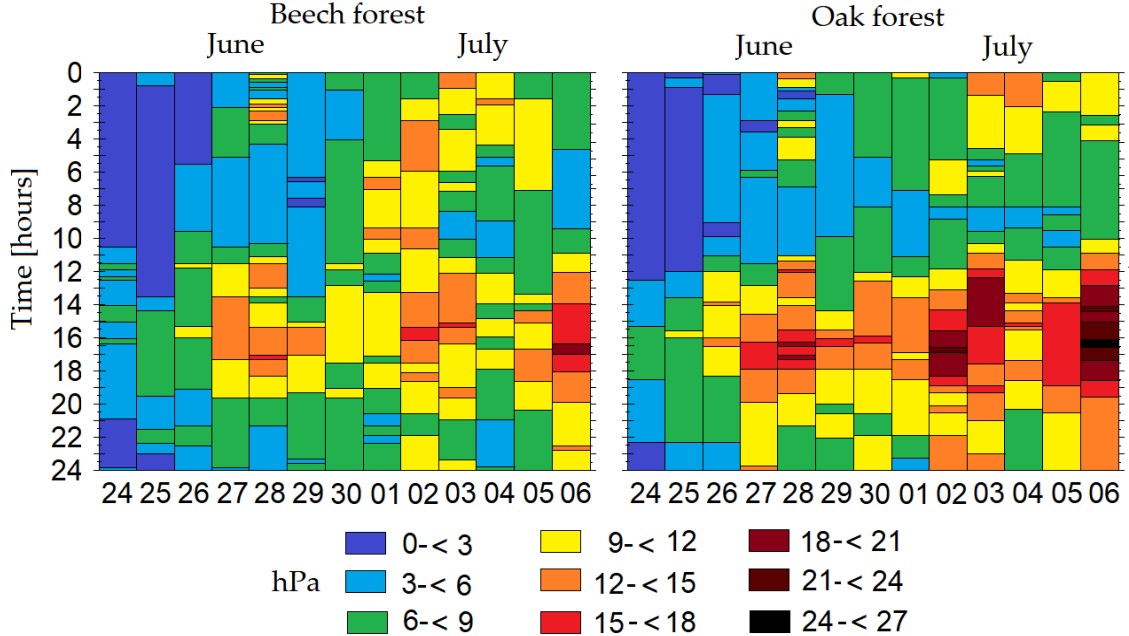

**Figure 5.** Diurnal variation in vpd on 13 days in summer 2019 (24 June—6 July) in the interior of the beech forest (**left**) and oak forest (**right**) of the Eşelniţa transect.

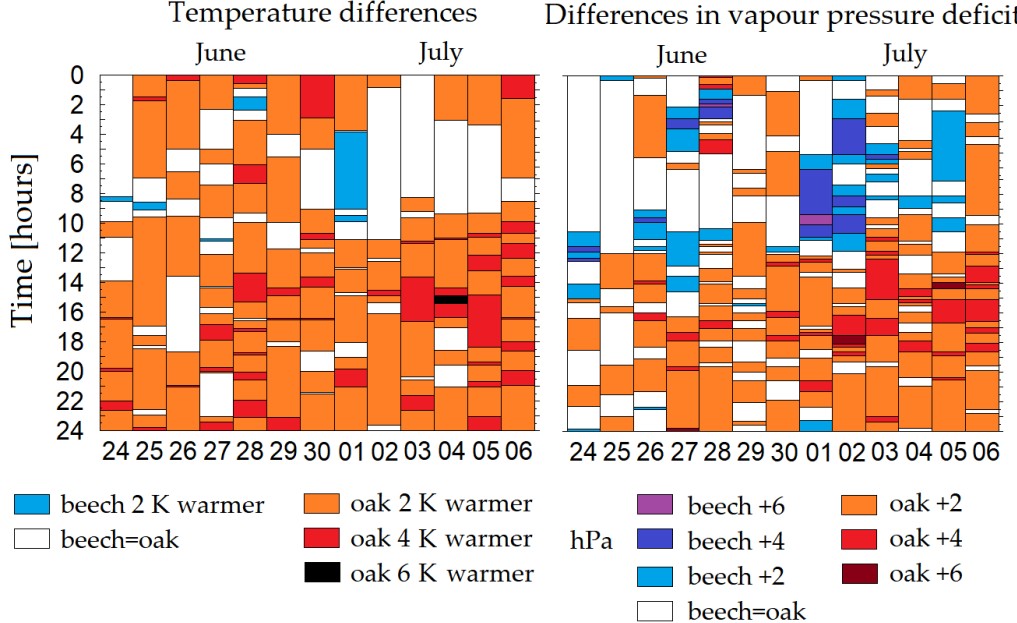

**Figure 6.** Differences in air temperature (**left**) and vpd (**right**) between the oak and beech forests of the Eşelniţa transect in their diurnal course over 13 days in summer 2019 (24 June–6 July). The vpd difference is given in hPa. Note that the oak forest interior is most of the day 2–6 K warmer than the beech forest, especially in the afternoon. vpd is higher in the oak forest especially in the afternoon and early night, while vpd is higher in the beech forest on several days late in the night and in the morning.

## 4. Discussion

Assuming a temperature lapse rate of c. 5 K km$^{-1}$ in western Romania, the average temperature difference recorded between the oak and beech forest interior of 1.2 K can be largely explained by the elevation difference of the two forest types (205–315 m), which should account for 1.0–1.6 K. This is corroborated by the average temperature difference found between the north-exposed beech forest at lower elevation, and the south-west exposed beech forest at higher elevation, of only 1.0 K, for which a corresponding temperature decrease with elevation of ≈1.3 K is calculated. However, temperature gradients between lower oak and higher beech forests are greater during daytime, both when considering means (2.0 K) and daily maxima (2.9 K). One possible explanation is a species-specific effect of the canopy on radiation transmission and the heating of the forest interior. Data from Central European beech and oak forests indicate that beech forests maintain higher leaf area indices (ca. 5.5–8.5) than oak forests (around 5.0) [4], which is well documented in the much higher PAR canopy transmissivity of oak canopies (about 10% of incident radiation) as compared to beech canopies (1–3%) [38]. Thus, more radiation is absorbed by the ground layer and the soil in oak forests, which leads to warming in the lower trunk space especially in the afternoon, while more radiation is absorbed in higher canopy strata in beech forests. The greater storage of heat on the ground and in the trunk space of the oak forests especially in the second half of the day (12 a.m.–midnight) is well demonstrated in the temperature-difference graph in Figure 5 and the much higher temperature maxima recorded in the oak forest (Appendix A Figure A1). During nighttime, more longwave radiation seems to leave the oak stand due to greater heat storage and the less dense canopy, resulting in colder temperatures in the second half of the night than in the beech forests. In the lower section of the oak/beech ecotone, where beech stems contribute with about 30%, thermal conditions are still more similar to the oak forest than to the beech forest. This type of mixed forest also has a quite high canopy transmissivity, though more patchily than in the pure oak forest.

Experimental evidence from saplings suggests that elevated vapor pressure deficits are a factor that can deteriorate the foliar water balance of beech and reduce growth, independently of soil moisture

status [22]. Mature beech trees show higher radial growth when vpd is low [39] and it is discussed that recent increases in vpd levels with rising temperature are one of the factors leading to vitality decline and increased mortality in temperate broadleaf trees, including beech [40]. Our data show a considerably higher average vpd in the oak than beech forests, when only the daytime values are considered (12.6 vs. 7.8 hPa); the difference is also large when vpd sums and vpd maxima are considered (Appendix A Figure A3). The 20% or even 34% higher vpd sum in the oak or oak-rich forests compared to the beech forests and the frequent exceeding of 15 or even 20 hPa may well mean that the atmospheric moisture conditions in the lower oak canopy are too stressful for beech to form its large and tender shade leaves. This could be one factor excluding beech from the sites with current oak dominance. Oak does not produce shade leaves with a similarly high SLA as beech [36] and thus likely is more tolerant of elevated vpd levels in the lower canopy. The fact that the north-exposed beech stands had lower vpd sums than the oak and oak-beech forests at similar elevation (90 vs. 104 and 126 MPa, difference not significant), may partly result from the northern exposition and lower radiation load, but it also suggests an effect of canopy structure on the air moisture regime, given that the vpd maxima were significantly lower in all beech stands. The considerably larger leaf area of beech especially in the middle and lower crown parts likely is reducing air turbulence more than in the more open oak forest canopies and this will contain moister air masses in the stand interior. This, together with the low radiation transmission, explains the relatively cool and moist stand microclimate described for beech forests by many vegetation ecologists [4,41–43]. The ability of beech to alter the stand microclimate toward a cooler and more humid environment can be considered as a form of climatological ecosystem engineering with consequences not only for the species' own shade foliage and offspring, but also for other biota on the forest floor. The vpd trends across forest types are similarly reflected by the relative air humidity data (Appendix A Figure A2); yet, the differences are less pronounced.

From the unexpected result that atmospheric moisture was in its long-term average higher, and not lower, in the oak forests at lower elevation compared to the ecotone higher upslope, we assume that nighttime values reduce vapor pressure and vpd differences between beech and oak forests, that develop during daytime. It is a well-known phenomenon that cooler and moister air masses frequently are moving downhill during night with the result that a temperature inversion can develop. When the moister and cooler air masses of the higher beech forests are moving downhill during the night and accumulate at the mountain base, relative air humidity might well be higher, and vpd lower, for part of the night in the oak forest than higher up the slope in the ecotone. Since nighttime conditions are physiologically less important, as long as stomates are closed, it is recommended to focus the microclimatological analysis on daytime means and extreme values.

Our microclimate measurements at 2 m height above ground inside the forests are of high relevance for the shade crown and the regeneration layer, but they do not allow conclusions on climatic stress experienced in the sun crown. Over-heating of the sun leaves over air temperature by 3K or more has often been reported for temperate broadleaf trees [44,45]. In the absence of temperature measurements at sun canopy height, we can use the measurements in the gaps to obtain a rough picture of the thermal stress experienced by the oak and beech trees in this mid-summer period. The gap stations (which are located at elevations between the oak and beech forest elevations) recorded average daily maxima of 30.7 °C for cloudless days, and absolute maxima exceeding 35.0 °C. Assuming that the sun leaves overheat in still air, one can assume that leaf temperatures near 40 °C or above are reached on certain summer days. The photosynthesis of beech has a broad temperature optimum; yet, the photosynthetic electron transport and $CO_2$ assimilation have been found to decline above 32–34 °C [20,21] and the heat sensitivity of leaf physiology is higher in beech than in other broadleaf tree species such as birch [20]. This suggests that beech could also be excluded from the warmer oak forest sites due to its greater heat sensitivity. Comparative physiological measurements in beech and oak canopies are needed to identify heat-induced limits of beech and oak.

Certainly, other climatic factors apart from heat and drought may limit the distribution of beech as well, notably winter cold, late frost events, and a short growing season. Our microclimatological study in the warmest months of the year is thus highlighting only part of the climatic constraints faced by beech. However, the study region in western Romania has a less continental climate than is experienced by *Fagus* at its eastern distribution limit in eastern Romania and Poland, where winter cold plays a decisive role [46]. This justifies the focus on summer heat and drought.

## 5. Conclusions

This gradient study shows that the microclimate in the forest interior is during mid-summer significantly warmer and drier in colline oak forests than in submontane beech forests with a difference in mean daytime temperature by ≈2 K and in mean daytime vpd of ≈2 hPa (questions 1 and 2). The difference in T and vpd was not much greater on cloudless days, but the absolute values were higher (question 3). During nighttime, the microclimate gradient is weaker or may even disappear, which could be due to downhill movement of cooler and moister air masses. Both the T and vpd gradients across the ecotone are larger than can be explained from the elevational temperature decrease, evidencing an effect of canopy structure and leaf area on the microclimate inside the stand (question 4). The by 2 K and 2 hPa warmer and more xeric microclimate inside the stand may well be one of the decisive factors that excludes beech from the sites currently colonized by oak forest. Our results can help to specify the thermal and hydrometeorological limits of beech more accurately. Even though nighttime temperature is relevant for respirative carbon losses, daytime temperature and air moisture extremes should largely determine the microclimatic limits of beech occurrence. Our T and vpd data point to thresholds in the range of 30 °C and 25 hPa in the lowermost shade canopy as maxima tolerated by beech for extended periods. The macroclimatic data derived for the three oak/beech ecotones further support the indicative value of an Ellenberg quotient of 30 for localizing the xeric limit of beech in Central and south-eastern Europe, in agreement with the suggestion of Ellenberg himself and subsequent authors.

**Author Contributions:** Conceptualization, S.H., C.L., A.I., and H.W.; Data curation, S.H.; Formal analysis, S.H.; Funding acquisition, H.W., A.I., C.L., and S.H.; Investigation, S.H.; Methodology, S.H. and C.L.; Project administration, H.W., A.I., C.L. and S.H.; Supervision, H.W., C.L. and A.I.; Validation, S.H. and C.L.; Visualization, S.H. and C.L.; Writing—original draft, S.H., C.L.; Writing—review and editing, C.L., S.H., H.W., and A.I. All authors have read and agreed to the published version of the manuscript.

**Funding:** The NEMKLIM project (Nemoral Forests under Climate Extremes) was jointly funded by the German Federal Agency for Nature Conservation (BfN) and German Federal Ministry for the Environment, Nature Conservation and Nuclear Safety, grant number 3517861300. A fellowship for S.H. was funded by the Transilvania University of Braşov, Romania.

**Acknowledgments:** We are grateful to M. Teodosiu, A. Petriţan, and D.-O. Turcu for assisting field work and V. Pacurar for exchanging ideas.

**Conflicts of Interest:** The authors declare no conflict of interest. The funders had no role in the design of the study; in the collection, analyses, or interpretation of data; in the writing of the manuscript, or in the decision to publish the results.

## Appendix A

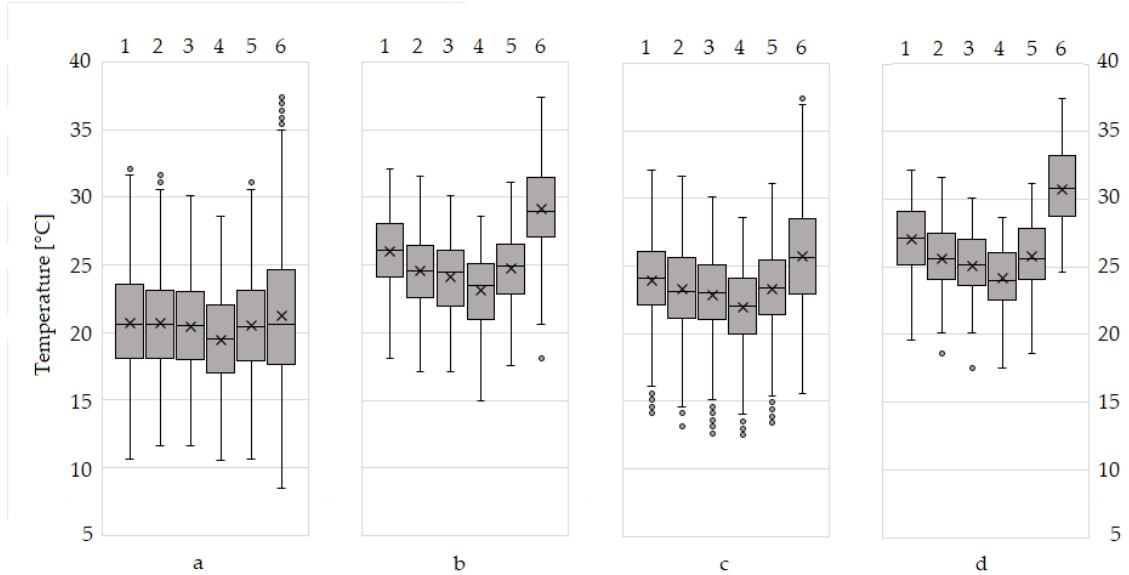

**Figure A1.** Boxplots with mean (x), median (horizontal line), 25 and 75 percentiles, and minima and maxima of air temperature in the study period at the six measuring points, given for all measured values (**a**), the recorded daytime maxima (**b**), the daytime means recorded in cloudless periods (**c**), and the daily maxima recorded in cloudless periods (**d**). The six measuring points are 1 = Oa, 2 = Oa/Be(lo), 3 = Oa/Be(up), 4 = Be, 5 = Be(N), and 6 = Ga.

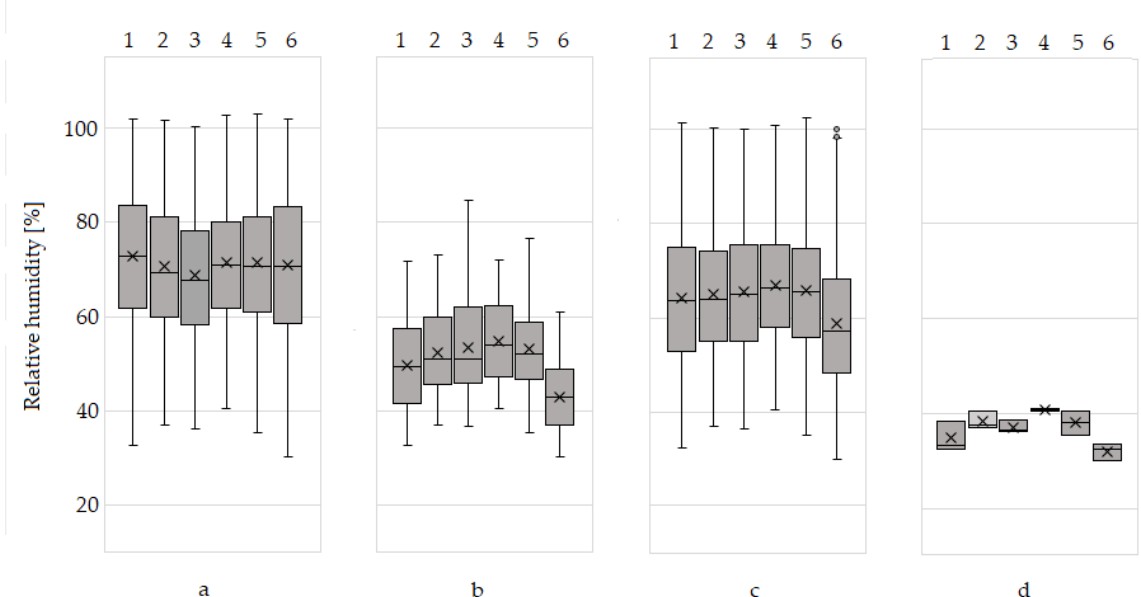

**Figure A2.** Boxplots with mean (x), median (horizontal line), 25 and 75 percentiles, and minima and maxima of relative air humidity in the study period at the six measuring points, given for all measured values (**a**), the recorded daily minima (**b**), the daytime means recorded in cloudless periods (**c**), and the absolute minima recorded (**d**). The six measuring points are 1 = Oa, 2 = Oa/Be(lo), 3 = Oa/Be(up), 4 = Be, 5 = Be(N), and 6 = Ga.

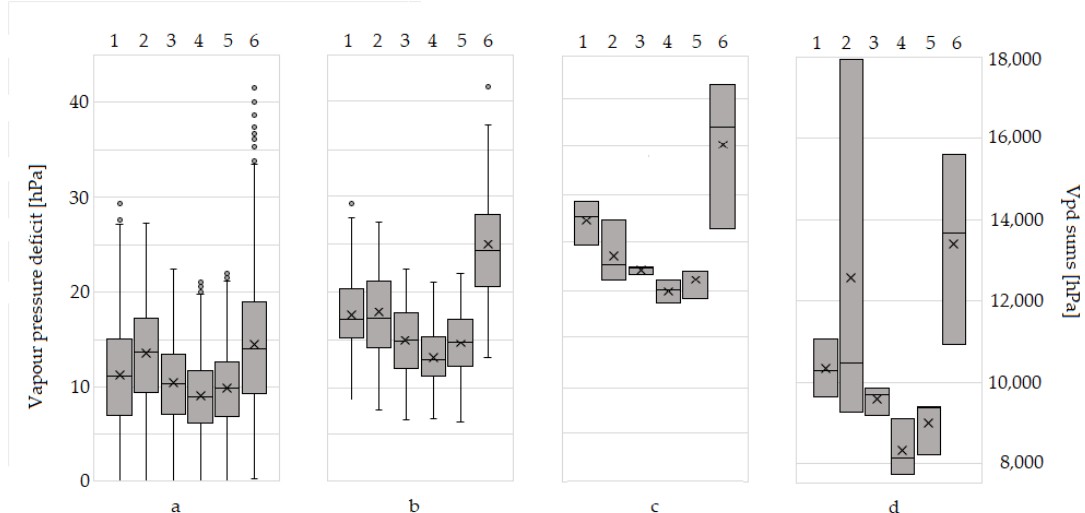

**Figure A3.** Boxplots with mean (x), median (horizontal line), 25 and 75 percentiles, and minima and maxima of vpd in the study period at the six measuring points, given for the daytime period on cloudless days (**a**), the recorded daily maxima in cloudless periods (**b**), the recorded absolute maxima (**c**), and the summed vpd values for the daytime period on cloudless days (**d**). The six measuring points are 1 = Oa, 2 = Oa/Be(lo), 3 = Oa/Be(up), 4 = Be, 5 = Be(N), and 6 = Ga.

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
