# Peer review of "Microclimatic Tipping Points at the Beech–Oak Ecotone in the Western Romanian Carpathians"

_forests, doi:10.3390/f11090919_

Round 1

Reviewer 1 Report

This study assessed the differences in summer air temperature, rel. humidity and vapor pressure deficit along three transects in a transitional oak to beech forest ecosystem in the Romanian western Carpathians.

The study is in the context of how climate warming may affect the shift of forest ecosystems on the landscape and suggests that European beech will likely experience range contraction in the future due to predicted climate warming.

I really enjoyed reading the study. It is well written and executed. I liked Figures 3, 4, 5 as these are nice visualization of daily temperatures changes over time (i.e. days).

The only thing the authors need to address, in my opinion, is how the statistical analysis has been conducted (line 190 to 195):

1) It appears to me that you conducted an analysis of variance using the aov() function in R to test whether at least one population mean is different with respect to the main effects and interaction of the underlying sample. Then, if the ANOVA was significant, you ran multiple mean comparisons (posthoc test) but did not mention which one you used and why, such as Tukey's HSD etc. Don't forget to mention the R function for this as well.

2) The normality assumption technically applies to the *residuals*, i.e. it will be checked *after* the model has been fit. More importantly however, is the assumption of equal variances, which has to be checked on the residuals as well. Those assumptions do not apply to the sample or raw data although one can often see potential problems in the raw data through initial data exploration.

I would highly recommend assessing those plots visually and not via statistical tests such as Shapiro-Wilk etc. since they are driven by sample size, e.g. large samples will be significant even if there is only a minor deviation from normality. Similarly, small samples are often not significant but certainly do not look normal. There are a lot of references for this on the web.

Here is a good reference: https://besjournals.onlinelibrary.wiley.com/doi/10.1111/j.2041-210X.2009.00001.x

3) A general comment, not directly directed to the authors: Model validations through the assessment of the assumptions of ANOVA et al. are often not performed well in my opinion and hence may cause biased parameter estimates and p-values. In order to convince me as a reader of the paper, I would always want to see the residual plots as well as boxplots of the raw data. Hence my recommendation would be to add them to the appendix. They don't need to be pretty and can simply be generated in R.

4) You also mentioned linear models (line 193). Were you referring to the ANOVAs there? Or did you fit additional linear models with continuous predictor variables (as opposed to categorical ones) using the lm() function?

Author Response

This study assessed the differences in summer air temperature, rel. humidity and vapor pressure deficit along three transects in a transitional oak to beech forest ecosystem in the Romanian western Carpathians.

The study is in the context of how climate warming may affect the shift of forest ecosystems on the landscape and suggests that European beech will likely experience range contraction in the future due to predicted climate warming.

I really enjoyed reading the study. It is well written and executed. I liked Figures 3, 4, 5 as these are nice visualization of daily temperatures changes over time (i.e. days).

The only thing the authors need to address, in my opinion, is how the statistical analysis has been conducted (line 190 to 195):

1) It appears to me that you conducted an analysis of variance using the aov() function in R to test whether at least one population mean is different with respect to the main effects and interaction of the underlying sample. Then, if the ANOVA was significant, you ran multiple mean comparisons (posthoc test) but did not mention which one you used and why, such as Tukey's HSD etc. Don't forget to mention the R function for this as well.

Thank you for this justified comment. We have adapted the description of the methods section accordingly. It reads now as follows:

Lines 205-215: We improved the text of the “methods” chapter as follows: Analysis of variance (ANOVA) was applied to test for significant differences between means of the five measuring points, using the aov() -procedure of the software platform R [37]. If at least one population mean was different from others with respect to the main effects and interaction of the underlying sample. Comparisons between all single means of the five different measuring points were conducted with ANOVA and subsequent multiple post-hoc testing, using the Tukey's HSD test, implemented of the aov() -procedure of R [35]. Linear models were calculated to examine the influence of the categorial variables measuring point (5 forest types and gap), transect (3 transects), and the measuring point x transect interaction on the means of temperature, RH and vpd, and their extremes, that were also considered in the ANOVA.

2) The normality assumption technically applies to the *residuals*, i.e. it will be checked *after* the model has been fit. More importantly however, is the assumption of equal variances, which has to be checked on the residuals as well. Those assumptions do not apply to the sample or raw data although one can often see potential problems in the raw data through initial data exploration.

Lines 212-219: This is correct and the methods description was changed (see above).

I would highly recommend assessing those plots visually and not via statistical tests such as Shapiro-Wilk etc. since they are driven by sample size, e.g. large samples will be significant even if there is only a minor deviation from normality. Similarly, small samples are often not significant but certainly do not look normal. There are a lot of references for this on the web.

Here is a good reference: https://besjournals.onlinelibrary.wiley.com/doi/10.1111/j.2041-210X.2009.00001.x

Lines 503-525: For illustration, some boxplots of the raw data are provided in the appendix.

3) A general comment, not directly directed to the authors: Model validations through the assessment of the assumptions of ANOVA et al. are often not performed well in my opinion and hence may cause biased parameter estimates and p-values. In order to convince me as a reader of the paper, I would always want to see the residual plots as well as boxplots of the raw data. Hence my recommendation would be to add them to the appendix. They don't need to be pretty and can simply be generated in R.

4) You also mentioned linear models (line 193). Were you referring to the ANOVAs there? Or did you fit additional linear models with continuous predictor variables (as opposed to categorical ones) using the lm() function?

Lines 205-219: The models used the categorial variables measuring point (forest type), transect and their interaction as predictor variables and different temperature, air humidity and vpd means/daily extremes as dependent variables. The latter variables were the same as tested with the ANOVAs.

Reviewer 2 Report

The topic of the paper Microclimatic Tipping Points at the Beech-Oak- Ecotone in the Western Romanian Carpathians” by  Stefan Hohnwald, Adrian Indreica, Helge Walentowski, and Christoph Leuschner seems to be actual and very interesting, but the text that I was able to download from the journal's website is not finalized as a finished article:

1) there is no the list of references

2) the appendix includes some illustration without any explanation. It is not clear why the authors decided to organize this section at all.

3) Lines 139-142 do not relate to the caption under the figure, meaning it should be a new paragraph.

4) The mention of Table 2 in the text comes before the mention of Table 1.

5) There is no any reference or derivation explaining how the formula (1) on Line 166 was obtained.

There are also some principle remarks:

1) The article describes the measurement methods and results very well. Based on the introduction, it seems that the goal of the authors is to get a picture of changes in the species composition of forests based on climatic trends, but the article does not contain any specific forecasts and proposals for Carpathian forests, at least within the framework of the model.

2) The authors should explain why it is possible to draw conclusions about the state of the forests based only on data on temperature fluctuations in the summer period, since it is obvious that the temperature regime in other seasons, especially in winter, certainly affects the change in the species composition of the studied forests.

Author Response

The topic of the paper Microclimatic Tipping Points at the Beech-Oak- Ecotone in the Western Romanian Carpathians” by Stefan Hohnwald, Adrian Indreica, Helge Walentowski, and Christoph Leuschner seems to be actual and very interesting, but the text that I was able to download from the journal's website is not finalized as a finished article:

1) there is no the list of references

To the editor: there must have occurred a technical problem because, of course, we provided a reference list (see manuscript)!

2) the appendix includes some illustration without any explanation. It is not clear why the authors decided to organize this section at all.

Lines 231-238: We incorporated the figure now into the text, including some explanations, as it seems to be proposed by the reviewer.

3) Lines 139-142 do not relate to the caption under the figure, meaning it should be a new paragraph.

Lines 129-139: Yes, unfortunately, there was a typesetting error, as the legend to Figure 1 continued directly in the normal text block. We corrected this.

4) The mention of Table 2 in the text comes before the mention of Table 1.

Lines 140 and 155: We placed now Table 2 before Table 1 and changed the numbers to solve the problem.

5) There is no any reference or derivation explaining how the formula (1) on Line 166 was obtained.

Lines 170-176: We explained now in detail, why we have obtained the Teten formula in the text, citing two more references.

There are also some principle remarks:

1) The article describes the measurement methods and results very well. Based on the introduction, it seems that the goal of the authors is to get a picture of changes in the species composition of forests based on climatic trends, but the article does not contain any specific forecasts and proposals for Carpathian forests, at least within the framework of the model.

Lines 415-416: We agree that many readers would expect conclusions from out measurements on future change in tree species composition due to a warming and drying of the climate. However, we feel that such conclusions are not justified from our microclimatic data set and they would represent an overextension of the information provided. There are certainly other factors besides microclimate that drive future vegetation changes, including the length of dry periods, the length of the vegetation period, and seed availability and regeneration success. Our data could be used in models that attempt predictions of climate-driven vegetation change, but this was not the scope of our strictly empirical study.

2) The authors should explain why it is possible to draw conclusions about the state of the forests based only on data on temperature fluctuations in the summer period, since it is obvious that the temperature regime in other seasons, especially in winter, certainly affects the change in the species composition of the studied forests.

Lines 416-420: Since the study region is located in the west of Romania with a less continental climate than in eastern Romania or eastern Poland at the continental distribution edge of European beech, winter cold and late frost in spring are less decisive than heat and drought in summer. This has been found in dendrochronological studies covering large parts of the beech distribution range, where low winter temperatures were found to be influential only at the eastern edges of the distribution or at higher elevation. We thus covered the likely most important microclimatic factors.

Round 2

Reviewer 2 Report

After making the corrections, the paper Microclimatic Tipping Points at the Beech-Oak- Ecotone in the Western Romanian Carpathians” by  Stefan Hohnwald et. al. got the finished form. I have only one small note: in the "Discussion" it is worth mentioning that the temperature, humidity and vpd graphs are included in the appendix. I think that the paper can be published in “Forests” after correcting this.

Author Response

We greatly appreciate your thoughtful comments that helped improve the manuscript. Thank you very much for your effort. We amended the section as suggested and referred to the temperature graph in line 341, to the vpd in line 354 and to the humidity graph in line 371.